# Amino Acids as the Potential Co-Former for Co-Crystal Development: A Review

**DOI:** 10.3390/molecules26113279

**Published:** 2021-05-28

**Authors:** Ilma Nugrahani, Maria Anabella Jessica

**Affiliations:** Pharmacochemistry Department, School of Pharmacy, Bandung Institute of Technology, Bandung 40132, Indonesia; marbel26.maj@gmail.com

**Keywords:** amino acids, zwitterionic, co-crystal, anionic co-crystal, ionic co-crystal, salt co-crystal, l-proline, solubility, bioavailability, chiral resolution.

## Abstract

Co-crystals are one of the most popular ways to modify the physicochemical properties of active pharmaceutical ingredients (API) without changing pharmacological activity through non-covalent interactions with one or more co-formers. A “green method” has recently prompted many researchers to develop solvent-free techniques or minimize solvents for arranging the eco-friendlier process of co-crystallization. Researchers have also been looking for less-risk co-formers that produce the desired API’s physicochemical properties. This review purposed to collect the report studies of amino acids as the safe co-former and explored their advantages. Structurally, amino acids are promising co-former candidates as they have functional groups that can form hydrogen bonds and increase stability through zwitterionic moieties, which support strong interactions. The co-crystals and deep eutectic solvent yielded from this natural compound have been proven to improve pharmaceutical performance. For example, l-glutamine could reduce the side effects of mesalamine through an acid-base stabilizing effect in the gastrointestinal fluid. In addition, some amino acids, especially l-proline, enhances API’s solubility and absorption in its natural deep eutectic solvent and co-crystals systems. Moreover, some ionic co-crystals of amino acids have also been designed to increase chiral resolution. Therefore, amino acids are safe potential co-formers, which are suitable for improving the physicochemical properties of API and prospective to be developed further in the dosage formula and solid-state syntheses.

## 1. Introduction

The drug’s physicochemical properties depend on the characteristics of its active pharmaceutical ingredients (API) and the type of formulation. Most API are formulated in solid forms, such as tablets and capsules, to be more economical, stable, and easier to administer to patients [1]. On the other hand, recently discovered drugs (around 60% to 70%) are considered to be BCS (Biopharmaceutics Classification System) class II (low solubility, high permeability) and BCS class IV (low solubility, low permeability). Poor solubility may lead to problems in bioavailability and therapeutic efficacy [2]. Accordingly, many researchers have been developing ways to modulate API physicochemical properties to enhance their therapeutic effectiveness [3]. One strategy to modify the dissolution rate is crystal structure modification by combining the API with a co-former to form co-amorphous systems, co-crystals, hydrates, solvates, and salts [4]. The improvement of the physicochemical properties depends on physicochemical characteristics of the API, and the co-former selected [5]. Recently, the co-crystal formation has become a field of interest because of its potential to modulate API physicochemical properties, such as solubility, chemical stability, and physical stability, thus affecting the shelf life of the drug and its therapeutic effects in the body [6].

Many co-crystallization techniques have been reported, including neat grinding (NG), liquid-assisted grinding (LAG), solvent evaporation (SE), gas anti-solvent precipitation (GAS), and many more [7]. Recently, the success of the “green method” for co-crystallization has prompted many researchers to seek other co-crystallization techniques that are more eco-friendly [8]. Mechanochemical reactions, including grinding without or with minimal solvents [9], have emerged as an efficient alternative for co-crystal synthesis due to their environmentally friendly process [10]. Researchers are also looking for the less dangerous co-formers with hydrogen bonding sites that can interact with the API’s functional groups, including amino acids [4]. Mainly amino acids are soluble and stable in water, thus facilitating the “green method” co-crystallization [11].

Functional groups that can form a supramolecular synthon, such as acid···acid, acid···pyridine, acid···amide, amide···amide, amide···pyridine, etc., are essential factors to form intermolecular hydrogen bonds in co-crystal synthesis [12]. Additionally, molecules that can form double hydrogen bonds are more likely to develop co-crystals [7]. The amino and carboxylic groups of amino acids are donor and acceptor groups [13], which tend to form hydrogen bonds with other groups, such as hydroxyl, carboxyl, pyridyl, and phenolic hydroxyl [14].

Recently, a new category of multicomponent crystal, namely ionic co-crystal, is garnering much attention due to its advantages, such as the simplicity and functionality to improve physicochemical properties of pharmaceuticals, food, fertilizers, and chiral resolution [15]. In that kind of co-crystal development, amino acids become the main co-former, i.e., in the ionic co-crystal formation of levodopa with LiCl and l-tyrosine and l-phenylalanine as the biological precursors [16]. The hydrated ionic co-crystals from enantiopure l-proline and racemic dl-proline with LiX (X=Cl, Br, I) were also reported can increase chiral resolution [17], as well as dl-amino acids alanine, valine, leucine, and isoleucine with LiCl [18].

Amino acids are also considered to be generally recognized as safe (GRAS), which means that they have low toxicity and are easy to find in natural products, such as wheat, rice, corn, etc. Thus, amino acids are a good choice and potential co-former for co-crystallization [19]. Based on their suitability, amino acids have recently been developed for many more pharmaceutical co-crystal arrangements. Here, this article dealt with reviewing the amino acid-based co-crystals and their opportunities. l-proline is discussed in depth due to it becoming the most used co-former with some advantages over the other amino acids [17,20,21,22].

## 2. Co-Crystals

Co-crystals were first discovered in 1844; however, the structure of co-crystals was only characterized in 1958, and the term co-crystal was first used in 1963 by Lawton and Lopez [1]. According to the Food and Drug Administration (FDA), co-crystals are defined as “multi-component solid crystalline supramolecular complexes composed of two or more components within the same crystal lattice wherein the components are in a neutral state and interact via non-ionic interactions” [23]. In general, co-crystals can be divided into two major groups, namely molecular co-crystals and ionic co-crystals. Molecular co-crystals contain two or more different neutral components and are sustained by hydrogen bonds or halogen bonds. In contrast, ionic co-crystals contain at least one ionic component and are supported by charge-assisted hydrogen bonds or coordination bonds if metal cations are present [24].

Co-crystals provide a new and effective way to modify API physicochemical properties while maintaining therapeutic activity [3]. This fact makes co-crystals more accessible than other methods, such as micronization [25], solid dispersion [26], salt formation [27], nanoparticle formation [28], and others. Important factors that need to be considered in changing the physicochemical properties of a drug without changing its pharmacological activities are the character of the API and the co-former used, the molecular interactions that occur, and the synthesis procedure [2]. In contrast to salt formation, which can only be applied to ionizable API and involve the transfer of hydrogen atoms between acidic and basic functional groups, the co-crystallization technique can be used for both ionizable and non-ionizable API. Co-crystals are also different from other solid forms, such as solvates and hydrates; a solvate is a solid containing an organic solvent molecule; a hydrate is a solid containing a water molecule [29]. In addition, a combination of cocrystal with hydrate or salt also can be occurred [22]. The illustration of solid-state types is depicted in Figure 1.

Generally, the co-crystal is made by combining the API with a suitable co-former to form a supramolecular synthon. There are two kinds of supramolecular synthon, namely homo-synthons (consisting of the same functional group) and hetero-synthons (consisting of different functional groups) [1]. Along with the times, the combination of multiple APIs into unit dose has become a popular drug development strategy, as well as in the development of co-crystal. A co-crystal formed from a mixture of APIs is called a drug–drug co-crystal (DDC) [20]. Recently, there have been many studies on DDC, such as co-crystal of tramadol hydrochloride-celecoxib, meloxicam-aspirin, sildenafil-aspirin, metformin-glipizide, and so on [1]. The ability of the API and co-former to form a co-crystal can be predicted by determining the ∆pKa value. When the ∆pKa of the API and co-former is negative (<0), there is no proton transfer, which indicates that the system will form a co-crystal; in contrast, when the ∆pKa is above 3, there is full proton transfer, indicating that the system forms a salt. Often, the ∆pKa is between 0 and 3, in which only a partial proton transfer occurs, so the system is commonly referred to as a salt co-crystal [7].

Various factors can influence co-crystallization success; one of them is the number of donor and acceptor groups of hydrogen bonds in the API and co-former [7]. In a co-crystal, the functional groups of the API and co-former interact via non-covalent interactions, including hydrogen bonds, van der Waals bonds, and π–π interactions [3]. Etter (1990) formulated hydrogen bond rules to predict the circumstances in which hydrogen bond interactions can form a co-crystal: (a) mostly all suitable proton donors (such as -COOH and -NH_4_^+^) and acceptors (such as -OH and -NH_3_) are utilized in hydrogen bonding, (b) six-membered ring intramolecular hydrogen bonds (such as C-H···O) are formed first in preference to intermolecular hydrogen bonds (such as N-H···O and O-H···O), (c) the best proton donors and acceptors available after intramolecular hydrogen bond formation then participate in intermolecular hydrogen bonds, and (d) all acidic hydrogen atoms are included in hydrogen bonding in the crystal structure [30]. In addition to the number of donor and acceptor hydrogen bond groups, the flexibility of the functional groups also plays a role in determining the success of co-crystallization [7].

Newly synthesized co-crystals must be characterized based on their crystal structure and physical properties to distinguish the co-crystal from the pure compound. Structural characterization involves powder X-ray diffraction (PXRD), single X-ray diffraction (SXRD), infra-red (IR) spectroscopy, and solid state-nuclear magnetic resonance (ss-NMR), while physical characterization involves melting point determination, differential scanning calorimetry (DSC), differential thermal analysis (DTA), thermogravimetric analysis (TGA), and scanning electron microscopy (SEM). Thermal analysis techniques, such as DSC, DTA, and TGA, are performed to assess the melting point, crystallization, sublimation, decomposition, and solid-state transition properties, as well as the amounts of volatile compounds. In contrast, microscopy techniques are performed to study solid-state properties, such as the crystal’s size, property, and surface characteristics [31].

Over the past few years, co-crystals have shown significant potential in drug development, especially in modifying the physicochemical and pharmacokinetic properties of API. Many studies have shown that co-crystal formation can increase the bioavailability of API in BCS class II, i.e., those with low solubility, and BCS class III, i.e., those with low permeability [1]. For API that have solubility problems, co-crystal formation is an effective method to increase solubility and the dissolution rate [32,33,34]. This property has prompted many researchers to develop co-crystallization techniques for non-steroidal anti-inflammatory drugs (NSAID), which mostly belong to BCS class II [35,36,37,38,39,40,41]. Due to their low solubility and slow dissolution rate, the maximum concentration is reached four to six hours after oral administration. However, NSAIDs require a fast onset of action [42]. That is why it is crucial to develop an effective method to modify the solubility properties of NSAID to achieve a rapid onset of action.

Meanwhile, for API that have permeability problems, the presence of a co-former in the co-crystal has been shown to change the polarity properties that affect drug permeability [43,44]. Although it is known for its ability to improve API solubility, co-crystal formation can also be used to decrease the solubility of API that have a short half-life [45,46,47]. Apart from the ability to increase the API bioavailability by increasing solubility or permeability, co-crystal formation has been shown to improve stability against moisture [48], heat [45,49,50], and exposure to light [1]. Thus, co-crystal formation could be an effective method to improve the physicochemical properties of API compared to the pure compound.

## 3. Amino Acids

Amino acids are essential components for the synthesis of proteins, enzymes, hormones, peptides, neurotransmitters, and other mediators, consisting of an amino group and a carboxylic group [51]. Twenty amino acids are found in many natural polypeptides [52] and can be classified based on their side chain and physiological function. Based on the side chain, amino acids can be divided into three groups, namely amino acids with non-polar and uncharged side chains, amino acids with polar and uncharged side chains, and amino acids with charged side chains (Table 1) [53].

Based on their physiological functions, amino acids are also divided into three groups, namely essential amino acids which cannot be synthesized in the body, non-essential amino acids which can be produced in the body, and semi-essential amino acids which can be produced in the body but in limited amounts [51]. Apart from being a constituent of proteins, several amino acids also play roles in regulating critical metabolic pathways needed to maintain health, growth, reproduction, and immunity. These amino acids are the functional amino acids, including l-arginine, cysteine, glutamine, leucine, proline, and tryptophan [54].

As amino acids consist of amino and carboxylic groups, amino acids can act as bases or acids by accepting or donating hydrogen ions. At a certain pH, which is called the isoelectric point (pI), the amino group in the amino acid is protonated so that it becomes positively charged (-NH_3_^+^). In contrast, the carboxylic group is deprotonated so that it becomes negatively charged (-COO^−^). It can be said that the amino acid is in the zwitterionic form (Figure 2) [52]. In aqueous solutions, the amino and carboxylic groups of amino acids will dissociate depending on the solution’s pH. Therefore, amino acids have at least two dissociation constants (pK), i.e., for the carboxylic group (pK_1_), the amino group (pK_2_), and the side-chain group, if applicable (pK_3_) [55]. Apart from glycine, all amino acids have at least one chiral center so that they are optically active, which means they can rotate plane-polarized light and have optical isomers, namely L and D [53].

Most amino acids can dissolve in water at ambient temperature with solubility ranging from the most soluble, such as proline, glycine, and alanine, to the least soluble, such as cysteine and tyrosine [55]. The addition of acids and bases can improve the solubility of amino acids that are less soluble due to salt formation between the acids or bases added to the amino acids. Additionally, the presence of other amino acids can increase their solubility in water. Most amino acids do not have good solubility in organic solvents, such as ethanol, due to the polarity factor of the amino acids. Of the 20 amino acids, proline shows good solubility in ethanol, while methionine, arginine, leucine, glutamic acid, phenylalanine, histidine, and tryptophan are slightly soluble in ethanol [53]. As shown by research on the solubility of amino acids in water, ethanol, and ethanol-water mixtures, the side chain of an amino acid has a significant effect on the solubility of the amino acid when ethanol is added to the solution. Amino acids with side chains containing a ring, such as proline, tryptophan, tyrosine, phenylalanine, and histidine, show a smaller decrease in solubility than aliphatic amino acids, amino acids with side chains other than rings, charged amino acids, and amino acids without side chains. This phenomenon is due to the ethanol-philic properties of the ring in the side chain, while the amino and carboxylic groups are ethanol-phobic. When more ethanol is added, it causes the water molecules to disrupt amino acid molecules with water molecules, thus causing a decrease in solubility [56].

Amino acids in water and buffer solution will remain stable at −80 °C for six months without degrading. Additionally, amino acids are generally stable in aqueous solutions at physiological pH and body temperature (37 °C), except for cysteine and glutamine [55]. Cysteine will undergo oxidation to cystine due to the presence of metal ions and the absence of a reducing agent. At the same time, the amide and carboxyl groups from glutamine will interact spontaneously and form pyroglutamate. At high temperatures and pressures, amino acid solutions remain stable, except for glutamine and asparagine, which will be degraded. The stability of glutamine and asparagine can be increased by forming a dipeptide, such as alanine-glutamine and leucine-asparagine. In a strong acid solution, glutamine and asparagine will be converted to glutamic acid and aspartic acid, and tryptophan will be degraded. In contrast, other amino acids remain stable and undergo only slight degradation. Conversely, in a strong alkaline solution, tryptophan remains stable, while other amino acids are degraded, especially at high temperatures [55].

Due to their good solubility properties, amino acids, especially proline, are widely used to make a mixture of natural deep eutectic solvents (NADES), a ‘green solvent’ that has recently been developed because it has the potential to replace organic solvents that generally have high toxicity and high volatility, thus releasing volatile organic compounds [57]. NADES is made by combining at least two natural compounds, such as organic acids, organic bases, amino acids, and sugars, which have hydrogen bond acceptor and donor groups [58]. As it is made from natural compounds, NADES is environmentally friendly, and extracts from NADES can be used directly in products without the need for any purification steps. Moreover, NADES has a melting point below the pure compound and far below the ambient temperature [59]. This property makes NADES, besides having high dissolving power, and being non-volatile at room temperature, chemically and thermally stable, and non-flammable [57]. Although NADES is often considered harmless, it is necessary to evaluate its toxicity and biodegradability to ensure the safety of NADES [60], as it was found that NADES combined with organic acids such as malonic acid exhibits a higher toxicity profile than sugar-based NADES [61].

Amino acid-based NADES has been shown to have the ability to dissolve polyphenols, such as rutin, compared to other natural compound-based NADES [57]. Research on phenol extracts from grape skins using NADES showed that NADES components were selected not only to enhance the solvent’s physicochemical characteristics, but also to increase the biological activity of the extract. An in vitro cytotoxic study showed that amino acid-based NADES is a good candidate for “green extraction” due to its low toxicity [62]. Recently, NADES research has provided an extraction solvent for bioactive compounds, a medium for enzymatic and chemical reactions, and a solvent for macromolecules such as lignins, polysaccharides, and water-insoluble compounds [59]. The applications of NADES are not limited to extraction solvents, but NADES is also being considered in the pharmaceutical field and nutraceutical formulations [63,64].

## 4. Amino Acids as Co-Formers

From the discovered large numbers of drugs, most of them have problems due to low solubility [2]. Therefore, several attempts were made to increase the dissolution of these drugs. One of these was the formation of amorphous solid dispersions (ASD) with water-soluble polymers, such as polyvinylpyrrolidone (PVP) and hydroxypropyl methylcellulose (HPMC). However, the formation of ASD often has problems with physical stability at relatively high humidity due to the hygroscopic nature of some of the polymers used [65]. The utilization of polymers leads to large bulk volume in tablets and capsules [56]. An alternative approach was taken to solve this problem, where the polymer was replaced by a co-former with a low molecular weight to form a co-amorphous system [66,67]. The usage of amino acids as co-formers for co-amorphous formation has been extensively studied. It has been shown to improve the stability and dissolution profile of API with solubility problems [68,69]. Amino acids can be used as co-formers because they have amino and carboxylic groups that can act as donors and acceptors of hydrogen bonds [70].

### 4.1. Screening for Amino Acids Used as Co-Formers

Initially, the amino acids used as co-formers were selected based on the amino acids present at the drug’s biological receptor binding site. This information promoted the assumption that amino acids in the receptor binding site can form a strong interaction with the API [66]. However, Laitinen et al. observed that the receptor amino acid was not a definite prerequisite for the formation of a strong interaction between API and amino acid. A strong interaction did not always occur between the API and the receptor amino acid, and non-receptor amino acids were also able to form strong interactions with the API [67]. Therefore, Kasten et al. conducted a study by mixing six API with 20 amino acids to determine the ability of the amino acids to form strong interactions with the API. This study concluded that non-polar amino acids, in general, are excellent co-formers [71], while acidic amino acids and polar amino acids are classified as poor co-formers [72]. Non-polar amino acids with cyclic groups, such as l-phenylalanine and l-tryptophan, and non-polar amino acids with a pyrrole group, such as l-proline, should be prioritized as first-choice co-formers due to their ability to provide superior dissolution rates [73].

Apart from being a co-former in co-amorphous formation, amino acids are also widely used as co-formers in co-crystal structure, especially with “green method” co-crystallization, which prompted many researchers to find a less dangerous co-former [74]. Shah et al. conducted a study to compare the effect of the co-former in co-crystal formation on the physicochemical properties of ritonavir, which has low water solubility. This study showed that amino acids can increase the solubility of ritonavir and can be used as an alternative co-former that can modify API physicochemical properties other than carboxylic acids [75]. Apart from being a natural compound and having a low risk, amino acids are mostly chiral compounds, so they are widely used in the pharmaceutical industry as potential co-formers to form chiral co-crystals and have the potential to create new subclasses of co-crystal due to their zwitterionic form, called zwitterionic co-crystals [76]. After doing a crystallographic search in the Cambridge Structural Database (CSD) with the help of SciFinder and the web-interactive Chemical Abstracts Service (CAS) database, it was found that the number of co-crystal structures involving natural amino acids is relatively high (Figure 3). Based on this study, there are no co-crystals that form with basic amino acids, such as l-histidine, l-lysine, and l-arginine; the most commonly found were co-crystals with non-polar amino acids, especially l-proline [77].

The formation of co-crystals between API with low solubility and amino acids has been shown to improve API solubility. One study was carried out by An et al., who found that the co-crystallization of febuxostat with pyroglutamic acid could increase solubility in various media, such as water and buffer (pH 1.2, pH 4, and pH 6.8) [12]. Shete et al. also found that itraconazole co-crystals with amino acids, i.e., aspartic acid, glycine, l-proline, and serine, besides being able to increase the solubility of itraconazole up to three-fold higher than pure itraconazole, also showed a wider zone of inhibition at lower concentrations, due to the increased permeability of itraconazole [78].

Moreover, the co-crystallization of an API with certain amino acid can also reduce the side effects of the API, such as that found in the co-crystal between mesalamine and glutamine. Mesalamine is an anti-inflammatory drug in the intestine, which, as with other anti-inflammatory drugs, has low bioavailability. Co-crystallization between mesalamine and glutamine led to greater bioavailability and a higher amount of mesalamine in the intestine because glutamine tends to accumulate in this area and can reduce the side effect of bloating because glutamine has an acid-base stabilizing effect in gastrointestinal fluids [79]. Co-crystals between API and amino acids have even been marketed, such as Steglatro^®^ (co-crystal between ertugliflozin and l-pyroglutamic acid) and Suglat^®^ (co-crystal between ipragliflozin and l-proline) [80].

### 4.2. l-Proline as the Most Fruitful Amino Acid Co-Former

As found in SciFinder, l-proline is the most common amino acid co-former [77]. Hence, it is discussed deeper in this sub-section. Tumanova et al. observed four APIs with different functional groups to determine their ability to form co-crystal with amino acids. The APIs include naproxen (containing a carboxyl group), levetiracetam (containing two amide groups), oxiracetam (containing two amide groups and a hydroxyl group), and diprophylline (a xanthine derivative, similar to caffeine and theophylline), but only naproxen was capable of forming a co-crystal with amino acids [76]. Next, they also developed naproxen-l-proline co-crystal as a potential zwitterionic co-crystal [81]. The same result was found by Nugrahani et al., who examined l-proline co-crystal formation with three API, i.e., mefenamic acid, ketoprofen, and diclofenac acid. Of the three API, only diclofenac acid was able to form a co-crystal with l-proline [82]. Losev and Boldyreva also studied co-crystal formation between an organic dicarboxylic acid, namely tartaric acid, and a group of amino acids containing a different carbon chain length between their amino group and carboxylic group. This study concluded that short-chain amino acids tend to form co-crystals, while long-chain amino acids tend to form salts [83]. This result was consistent with other research on the effect of carbon chain length from the dicarboxylic acid, which is the most commonly used co-former for co-crystallization. Dicarboxylic acids with long carbon chains are not suitable candidates for co-crystallization due to the poor geometric suitability to the API [7]. These studies revealed that the formation of co-crystals depends on the selected API and the co-former.

A co-crystal is a homogenous solid phase containing two or more molecules in a static lattice with a specific stoichiometry, joined by weak interactions, especially hydrogen bonds. Therefore, the miscibility of the API and the co-former is a crucial parameter in the formation of a co-crystal. The miscibility between the API and the co-former can be predicted by calculating the solubility parameter (δ). When the ∆δ value between API and co-former is below 5, the system forms a co-crystal [29]. This method was developed by Shete et al. in their study on the formation of co-crystals between itraconazole and amino acids. Of the five selected amino acids, only glycine and l-proline produced a ∆δ value below 5. However, after being synthesized and characterized by FT-IR, DSC, and PXRD, the five amino acids were able to form co-crystals with itraconazole [78].

The selection of amino acids as co-formers for co-crystal formation can also be conducted by calculating the ∆pKa value, as described in Section 2. This approach is the most frequently used method for co-former selection because it is easy and does not take much time. In their study on the formation of febuxostat co-crystals with amino acids, An et al. used this method to determine the ability amino acids to form co-crystals. Of all the amino acids, those containing hydrophilic NH, O, and OH groups were found to have ∆pKa value below 3. Thus, five amino acids were found to be potential co-former candidates for febuxostat. However, not all amino acids that have ∆pKa value below 3 can form a co-crystal. Only one of the five amino acids could develop co-crystal with febuxostat [12]. Similar results were also found in a study conducted by Othman et al., who found that although l-alanine and l-proline have ∆pKa values below 3, only l-proline could form co-crystals with ibuprofen [20].

Therefore, a more reliable method for co-former selection was explored, such as using DSC thermograms to predict new phase formations. The thermogram of a physical mixture shows two endothermic peaks, each corresponding to the pure compound’s melting point, while a co-crystal thermogram shows one endothermic peak [7]. Moreover, melting point of a co-crystal is usually between the melting points of the pure compounds (about 54.6%); however, the melting point of the co-crystal can also be below (30.6%) and above (14.8%) that of the pure compounds [84]. Typically, the data obtained from DSC analysis are used to construct binary phase diagrams. The initial endothermic temperature on the DSC thermogram is generally chosen as the solid point. In contrast, the second endotherm’s temperature is chosen as the liquid point to form the phase diagram. The form of the new phase can be distinguished based on the shape of the phase diagram (Figure 4) [7]. Nugrahani et al. used thermal analysis through the construction of the melting point versus the composition of diclofenac sodium/potassium hydrate and l-proline to determine which ratio could form a salt co-crystal [22,85]. The co-crystal will compose a ‘W’ shape on the phase diagram with two eutectic points and one higher melting point between them as shown in Figure 4c [7,22,85].

However, besides considering the API and the co-former to be chosen, the type of solvent used also contributes to the success of co-crystallization. The polarity of the solvent determines the kind of non-covalent interactions and thus affects the intermolecular interactions in the co-crystal. Co-crystals with hydrogen bonds are more common in slightly polar solvents, while solvents that are more polar tend to form co-crystals with halogen bonds [7].

Tumanova et al. conducted a study on the ability of co-crystal formation between flurbiprofen and l-proline using the LAG method with five different solvents, i.e., methanol, ethanol, isopropanol, acetonitrile, and water; both compounds are soluble in methanol and ethanol; only flurbiprofen is soluble in isopropanol and acetonitrile, and only l-proline is soluble in water. This study showed that the choice of the solvent used for the LAG method is a factor that can affect the rate of reaction and the reaction pathway, as five co-crystal structures were found with different types of supramolecular synthon [86]. A similar thing was observed in the formation of co-crystals between naproxen and l-proline using the LAG method with five different solvents by Tilborg et al. [87].

#### 4.2.1. The Structure of l-Proline-Based Co-Crystals

In crystalline form, amino acids are zwitterions. They tend to form a head-to-tail charge-assisted hydrogen-bonded chain, which is also observed in the amino acid co-crystal, thereby increasing the percentage of strong interactions in the co-crystal structure [76]. In their research on the formation of zwitterionic co-crystals between naproxen and l-proline, Tilborg et al. found that l-proline forms two types of hydrogen bonds with carboxylate groups and protonated nitrogen, allowing the carboxyl group of naproxen to bind to the l-proline molecule. l-proline forms adjacent columns of l-proline units, and each l-proline molecule linked to the naproxen molecule by charge-assisted hydrogen bonding (Figure 5a) [87], which is also found in the l-proline co-crystal with fumaric acid (Figure 5b) [21]. The research concluded that the formation of a zwitterionic co-crystal structure supports the establishment of hetero-synthon hydrogen bonds, which is preferable in co-crystals [21,87]. Likewise, in flurbiprofen co-crystals with l-proline, the amino acid molecule forms two O-H···O interactions, one with R-flurbiprofen using O and involved in the l-proline-l-proline binding and one with S-flurbiprofen including O from other carboxylate groups. Flurbiprofen has a strong tendency to form racemic compounds through direct binding. Therefore, in co-crystal form, flurbiprofen is present in the form of R and S. When l-proline is added, the structure changes to a ‘sandwich-like’ trimer, like the zwitterionic co-crystal structure described in another study (Figure 5c) [88].

Organic molecules include amino acids and can form ionic co-crystal with an ionic salt. The ionic co-crystal, which contains alkali halides, was commonly classified as a coordination polymer. Meanwhile, an ionic co-crystal that consists of an organic cation halide is classified as a salt co-crystal. Most of the ionic co-crystal consists of organic cation halides because of their properties as suitable hydrogen bonding acceptors and can form intermolecular interactions with additional neutral molecules [1]. The co-crystal between diclofenac sodium and l-proline discovered by Nugrahani et al. is an example of amino acid salt co-crystals [22]. The salt co-crystals asymmetric unit consisted of l-proline, which coordinated with the alkaline ion and interacted with water and diclofenac molecules via hydrogen bonding, as shown in Figure 6 [22]. The salt co-crystal enhanced the alkaline salt solubility more than threefold.

Nugrahani et al. also developed a similar salt co-crystal between potassium diclofenac and l-proline, which had two pseudopolymorphs: monohydrate and tetrahydrate forms. As well as Na^+^ in sodium diclofenac-l-proline salt co-crystal, the K^+^ was a hexa-coordinated structure by two axial water molecules, two connecting water, and two oxygen atoms from the l-proline’s carboxylate group. The presence of water molecules in the hydrate co-crystal structure strengthened each l-proline hydrogen bond, thereby enhancing the interactions in the co-crystal [85]. These structural features likely improved the physicochemical properties of the API, particularly in dealing with unstable API, such as increasing the stability of highly hygroscopic lactic acid by co-crystallization with tryptophan [89].

From recently discovered co-crystals with amino acids, API co-crystals with l-proline are the most studied due to their ability to co-crystallize with various components. This compound is an excellent candidate in co-crystal formation because it forms α-ammonium carboxylate zwitterions, which support atomic interactions [20]. l-proline has a reasonably broad zwitterion pH range—1.8 to 10.63 [21]. In contrast with other amino acids, only l-proline contains a five-member ring structure, the pyrrolidine ring, making the l-proline structure rather rigid. In co-crystal formation, this stiffness is an advantage over the more flexible co-formers [22]. The presence of pyrrolidine groups in L-proline contributes to its good stabilization ability [73]. Hence, l-proline is a stabilizer for both non-salt mixtures such as co-amorphous systems [66] and ionic/salt co-crystal [17,22,85] by increasing the interaction between components.

Ionic co-crystals of LiX (X=Cl, Br, I) with l-proline and dl-proline were reported. Those solid phases consisted of the inorganic element with this amino acid formed conglomerates (with Cl and Br) and racemates (with Cl and I), which produced distinct crystal layers between the opposite chirality. Hence, this developed method offers an advantage to chiral resolution [17].

#### 4.2.2. Modification of the Physicochemical Properties of l-Proline-Based Co-Crystal

l-proline is the most soluble amino acid and can dissolve hydrophobic molecules with a hydrotropic effect. In addition, l-proline also supports the permeability properties of API, i.e., indomethacin [90]. This amino acid builds a flexible but intense interaction with another compound based on a zwitterionic interaction and a rigid structure, which added value for this co-former. All these things mean that l-proline is adequate to develop co-crystal forms with API with both solubility and stability problems.

Co-crystallization with the amino acid l-proline has been widely applied to increase the solubility of NSAID, mostly with low solubilities, such as naproxen [81,86,91], diclofenac acid [92], and ibuprofen [20], whose zwitterionic co-crystal structure has been investigated. It showed the ability to form co-crystals with several amino acids, such as alanine, tryptophan, tyrosine [20], and arginine [93,94]. Apart from naproxen, the formation of co-crystals between ibuprofen and l-proline was assessed by FT-IR and PXRD analysis showing a peak shift, which indicates that a new phase was formed when co-crystallization was carried out using the LAG and NG methods [20]. A comparison between the two co-crystallization methods, LAG and NG, was performed by Nugrahani et al. in co-crystal formation with diclofenac acid and l-proline. From these observations, it was found that the specific peak transformation of the co-crystal became more evident as the co-crystallization time increased, while the peak of the pure compound decreased. Co-crystallization with the NG method showed a similar FT-IR spectral pattern as the LAG method; however, the initial co-crystal was formed faster in the LAG method [20]. This phenomenon is consistent with other studies showing that that the addition of solvents can facilitate molecular diffusion, thereby increasing the interaction between API and co-formers [72].

Diclofenac acid-l-proline co-crystals have been shown to increase the solubility up to 7.69 times above that of the pure compound. The co-crystal was relatively stable at 30 °C/75% relative humidity and did not show changes in the diffraction pattern and peak intensity in the PXRD analysis [92]. After being compared with diclofenac sodium salt, the most common form of diclofenac acid on the market, it was found that the increased solubility of the diclofenac acid-l-proline co-crystal was still lower than that of diclofenac sodium. However, diclofenac sodium on the market is mostly a mix of the hydrated and anhydrous forms, wherein the hydrated form can return to the anhydrous form by heating and drying [95]. Therefore, a co-crystal between sodium diclofenac hydrate and l-proline was made to improve this stability problem. From the characterization results, it was found that the co-crystal is a tetrahydrate and was stable under high humidity conditions. This co-crystal also showed better solubility and a faster dissolution rate than diclofenac sodium [21]. The same effect was observed in the co-crystal between diclofenac potassium and l-proline. The co-crystal showed a higher solubility, about 3.56 times higher, and an accelerated dissolution rate, approximately 3.36 times faster than diclofenac potassium [88].

Co-crystals between NSAID and l-proline can also increase the permeability of the API, as studied by Wang et al. in their study on zwitterionic co-crystal formation with indomethacin and l-proline. The co-crystal form can increase solubility up to two to three times higher than pure indomethacin. This result is due to the breakdown of the dimer formed between carboxylate groups in the indomethacin molecule, which is the cause of its improved solubility with l-proline. The co-crystal form also showed a faster dissolution rate in media with various pH. It was concluded that the intrinsic dissolution rate (IDR) increases with increasing pH with an IDR value two times higher than that of the pure compound at all pH tested. From the permeability study, the co-crystal flux pattern was similar to the pure compound, but the co-crystal showed a significant increase within 1 h with higher concentrations. Additionally, the cumulative diffusion amount of indomethacin in the co-crystal form was 1.66 times higher than that of pure indomethacin. From the pharmacokinetic study, the co-crystal showed a maximum concentration that was similar to that of pure indomethacin. However, it was reached in a shorter time, and the level of the co-crystal was higher than the pure compound during the elimination process [90].

Increases in solubility and permeability were also observed in co-crystals between acetazolamide, an anhydrase inhibitor for lowering intra-ocular pressure that belongs to BCS class IV, and l-proline. From the dissolution study, all the acetazolamide-l-proline co-crystals in buffer media with different pH showed a similar dissolution profile, but the maximum concentration was reached in a faster time and was maintained under steady-state conditions until the end of the test. The dissolution profile also showed a significantly higher level than the pure compound. The time to reach the maximum concentration decreased with increasing pH, and the concentration achieved increased with increasing pH. Calculation of the IDR value showed that the IDR value for the co-crystal was three to four times higher than that of pure acetazolamide. Similar to itraconazole, acetazolamide molecules tend to bind with each other to form strong hydrogen bonds, which results in poor dissolution properties. In the co-crystal, the presence of l-proline breaks the hydrogen bonds of acetazolamide aggregation, resulting in better dissolution profiles. From the permeability study, the co-crystal showed a higher cumulative diffused amount than the pure compound, indicating that the co-crystal had better permeability properties. Additionally, the pharmacokinetic profile of the co-crystal showed a higher maximum concentration (Cmax) and area under the curve (AUC), a faster maximum time (Tmax), and a longer half-time (T1/2) than pure acetazolamide. This suggests that the co-crystal has better bioavailability than the pure compound [68]. A similar effect was observed with chlorothiazide, a diuretic drug that also belongs to BCS class IV. Co-crystallization between chlorothiazide and l-proline showed an improvement in dissolution properties compared to pure chlorothiazide [96].

Several amino acids, including l-proline, have also been used to form zwitterionic co-crystals with a 2:1 ratio with an inorganic lithium salt. Lithium is a monovalent metal, so it is challenging to develop a co-crystal in a 2:1 molar ratio. Therefore, a carboxylic group in the amino acid zwitterion is used to act as a link between two lithium cations to form a zwitterionic co-crystal of the amino acid-inorganic lithium salt [11]. Lithium salts, such as lithium carbonate and lithium citrate, are often used as mood stabilizers for bipolar therapy. However, this compound has a narrow therapeutic window and requires regular monitoring of plasma lithium levels to reduce its side effects. Therefore, researchers have looked for other anions that could replace carbonate and citrate but showed lower toxicity. Stolk et al., in a pharmaco-epidemiological study, found that acetylsalicylic acid (aspirin) was useful as an adjunct treatment with lithium salts [97], so they used salicylic acid, which is the primary bioactive metabolite of aspirin, as an anion pair to form a salt with lithium to produce lithium salicylate. Then, a co-crystal was made of lithium salicylate and L-proline. The pharmacokinetic profile exhibited by this co-crystal was very similar to the controlled-release formulation of lithium which has been approved by the FDA; however, this co-crystal had a better safety profile, as the 4 mEq/kg dose in mice provided a consistent increase in the plasma level of lithium for up to 48 h, while lithium carbonate was not even detectable at 48 h and produced a large spike at 24 h after administration [98]. A study on the safety, pharmacokinetics, and therapeutic efficacy of the lithium salicylate-l-proline co-crystal showed that the co-crystal could reduce β-amyloid plaques and phosphorylation by reducing nerve inflammation and inactivating 3-β-glycogen synthase kinase [99]. A comparative study of the prophylactic effects of the lithium salicylate-l-proline co-crystal, lithium salicylate, and lithium carbonate showed that, at low doses, they could prevent spatial cognitive decline and depression-like behavior; however, the co-crystal showed an advantage in preventing associative memory decline and irritability in mice [100].

Not limited to NSAID, diuretic, and epilepsy drugs, co-crystallization with l-proline has been studied for the lipophilic hypo-cholesterol agent ezetimibe. After characterization with FT-IR, Raman spectroscopy, DSC, TGA, and PXRD, the dissolution study showed an increasing concentration, five times higher in the co-crystal form within 3 min. After 2 h, the concentration of ezetimibe in the co-crystal decreased, but was still higher than that of pure ezetimibe. The co-crystal also showed good stability after one month of storage at 40 °C/75% RH [101]. The co-formers usually exhibited a dissolution phenomenon called “spring and parachute”, meaning that the release of the API to the highest supersaturation level occurred quickly, followed by precipitation inhibition in the dissolution medium for the desired time (Figure 7). The driving force for supersaturation (spring) is the difference between the amorphous and crystalline free energy [65].

In contrast, the period for maintaining supersaturation (parachute) reflects the system’s ability to inhibit deposition [65]. This phenomenon plays an essential role in increasing the solubility and dissolution rate of the co-crystal [3]. The explanation for the “spring and parachute” phenomenon was that the co-crystal dissociates to amorphous or nanocrystalline drug clusters during the supersaturation (the spring), which transform via fast dissolving metastable polymorphs to the insoluble crystalline modification to give high apparent solubility for co-crystals and optimal drug concentration (the parachute) in the aqueous medium [102]. The presence of the second component in the binary mixture generally reduces the free energy difference between the amorphous and crystalline forms, thereby acting to reduce the driving force for dissolution [65].

The spring and parachute phenomenon is also seen in the co-crystal between l-proline and flavonoids, which are natural polyphenol compounds with antioxidant, antitumor, and anti-inflammatory activities, but with low solubility and bioavailability. The flavonoids tested included flavonols (quercetin and kaempferol), flavones (luteolin, baicalein, and chrysin), and isoflavones (genistein). The phenolic hydroxyl group was preferred to form the charge-assisted hydrogen-bond with the carboxylate from the CSD survey. On the other hand, flavonoids have many phenolic hydroxyl groups, and l-proline is in the form of zwitterionic under physiological conditions. It was estimated that supramolecular hetero-synthons were formed between the phenolic hydroxyl groups and carboxylate groups. The dissolution study found that several co-crystals showed significant differences from the pure compound, such as chrysin-l-proline, baicalein-l-proline, and kaempferol-l-proline. In general, this higher solubility was maintained for a short time, then decreased rapidly, as described in the spring and parachute phenomenon. Only the kaempferol-l-proline co-crystal could support a final concentration 1.8 times higher than that of pure kaempferol. Therefore, a pharmacokinetics study was carried out on the kaempferol-l-proline co-crystal, where the test results showed that the co-crystal had a different curve shape with a shorter Tmax, higher Cmax, and larger AUC than that of pure kaempferol and physical mixture (Figure 8) [103].

Liu et al. also researched the formation of co-crystals between myricetin, a flavonoid, and l-proline. As with other classes of flavonoids, myricetin has low solubility and bioavailability. Myricetin has six hydroxyl groups and one carbonyl group as donor and acceptor hydrogen bond groups, making it possible to form co-crystals with the appropriate co-former. Co-crystallization myricetin with l-proline produced a co-crystal with higher solubility and a faster dissolution rate than the API itself. The solubility of the co-crystal was 7.69 times higher in 40 min than the parent compound. Within 10 min, the concentration of myricetin in the co-crystal was 6.28 times higher than myricetin in the physical mixture. As with the co-crystal between flavonoid and l-proline, the spring and parachute phenomenon also occurred with the myricetin-l-proline co-crystal; however, at the end of the test, the concentration of myricetin in the co-crystal was still higher than that of pure myricetin or myricetin in the physical mixtures. From the pharmacokinetic study, the Cmax and AUC of myricetin in the co-crystal were much higher than that of pure myricetin. The co-crystal also exhibited a faster Tmax and higher relative bioavailability than the parent compound. These results suggest that co-crystallization between myricetin and l-proline is an effective method to increase the plasma concentration and prolong myricetin’s absorption time [15]. Determination of the dissolution rate of the myricetin-l-proline co-crystal in media with various pH and tween-80 containing medium showed the highest IDR value in pH 4.5 buffer medium, while in the tween-80 medium, the IDR value of the co-crystal was lower than that of pure myricetin. This fact was related to the interface barrier formed by tween-80 due to its large molecular size and shape, thus inhibiting myricetin molecules from entering the bulk solution [104].

Báthori and Kilinkissa investigated whether γ amino acids, such as baclofen, can be used as a co-former to form a multi-component crystal. Baclofen is a hydrophobic γ amino acid which is slightly soluble in water. The γ amino acids, although rarely used, have similar hydrogen bonding properties to α amino acids. Therefore, γ amino acids can be co-crystallized with the appropriate co-former. Adequate molecular flexibility can accommodate co-former molecules, even with varying molecular sizes. The characterization of the crystal structure, thermal analysis, and PXRD analysis demonstrated the formation of co-crystals between baclofen and benzoic acid. This study concluded that structural flexibility and strong hydrogen bonds are essential for crystal engineering [105], which was also proven in the nano-crystallization of diclofenac acid-l-proline [106].

## 5. Brief Review

Finally, as discussed above, co-crystal development using amino acids as co-formers has been widely developed. Aside from being the potential co-former that improved physicochemical properties of the API, amino acids are also less toxic than chemical co-formers such as dicarboxylic acid, and easy to handle. Hereafter, from the regular size to nanoparticles of pharmaceutical-amino-acid co-crystal is waiting to be developed further. Table 2 presents the summary of a few literature reports available on co-crystals between API and amino acid.

## 6. Conclusions

Amino acids are potential co-formers for supporting the “green method” co-crystallization of API. Co-crystals formed by API and amino acid have been shown to improve the drug’s physicochemical properties, such as solubility, dissolution rate, bioavailability, as well as chemical and physical stability. Structurally, amino acids in a co-crystal are in their zwitterionic form and produce a head-to-tail charge-assisted hydrogen-bonded chain with the API, building strong interactions and enhancing stability. l-proline exhibits the highest capacity to constructs co-crystal and modulates the pharmaceutical performance due to its rigid zwitterionic structure, solubility, and hydrotropic activity. In addition, some ionic co-crystals from amino acids include l-proline were also developed to separate their chiral mixture. Hereafter, amino acid co-formers, especially l-proline, can be explored further for superior dosage forms development and chiral resolution methods in a simple, safe, economist, and eco-friendly way.

## Figures and Tables

**Figure 1 molecules-26-03279-f001:**
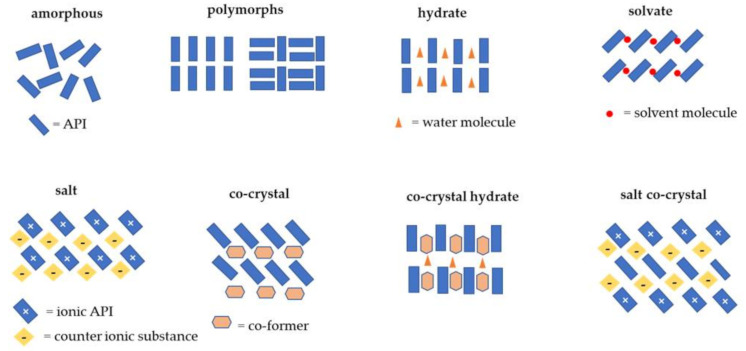
Illustration of the difference between co-crystal and other solid forms [22,29].

**Figure 2 molecules-26-03279-f002:**
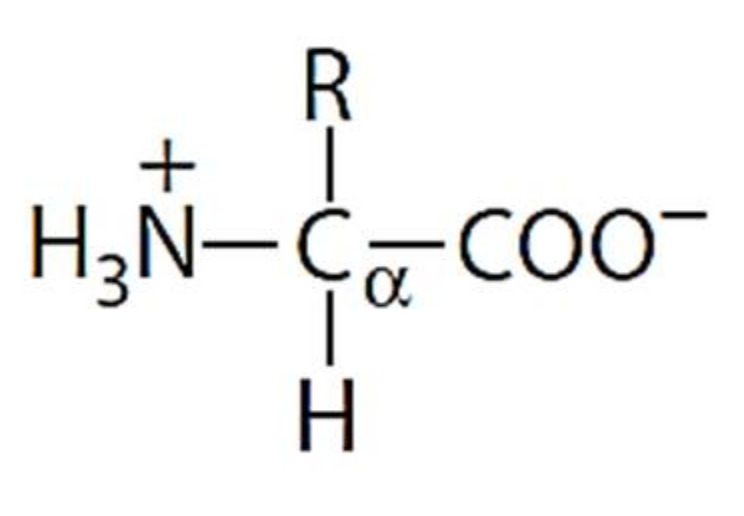
The zwitterionic (ionized) form of an amino acid [52].

**Figure 3 molecules-26-03279-f003:**
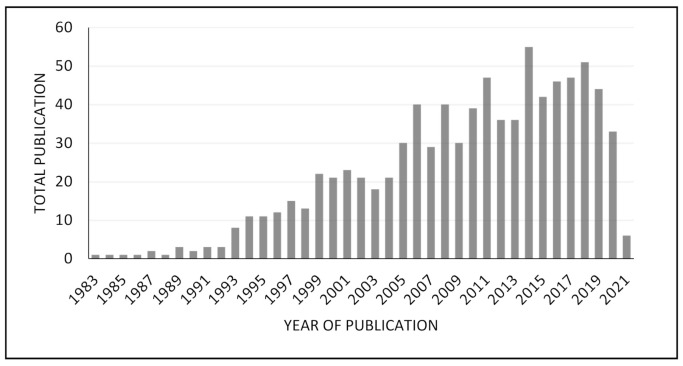
Bibliographic research in Pubmed for amino acid co-crystals [77].

**Figure 4 molecules-26-03279-f004:**
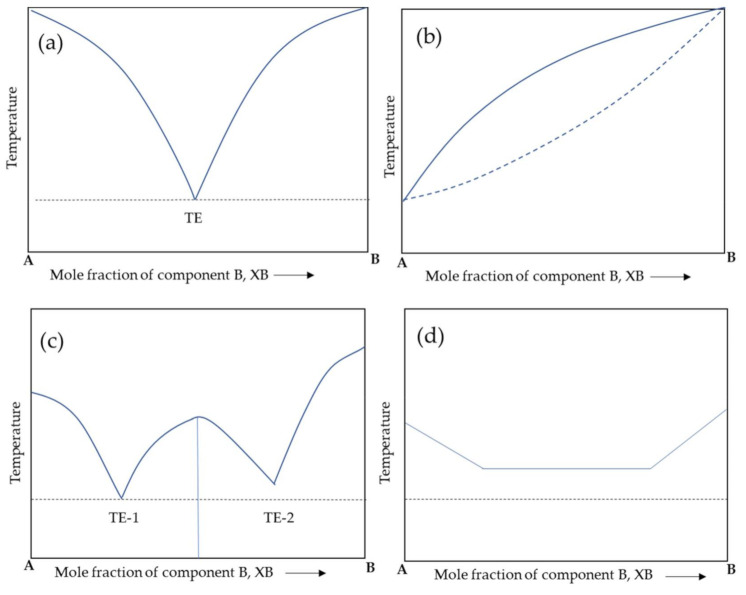
Schematic representation of binary phase diagrams for eutectic formation (**a**), solid solution formation (**b**), co-crystal formation (**c**), and physical mixture (dotted line—onset of first component melting; dark line—melting of the second component) (**d**). (Reproduced from [7]). Note: TE = temperature of eutectic point.

**Figure 5 molecules-26-03279-f005:**
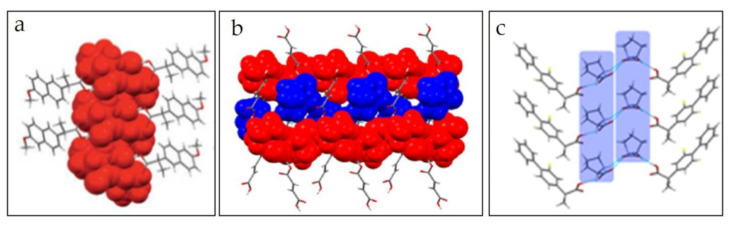
Single crystal structure of S-naproxen-l-proline co-crystal [87] (**a**), fumaric acid-l-proline co-crystal [21] (**b**), and RS-flurbiprofen-l-proline co-crystal [88] (**c**).

**Figure 6 molecules-26-03279-f006:**
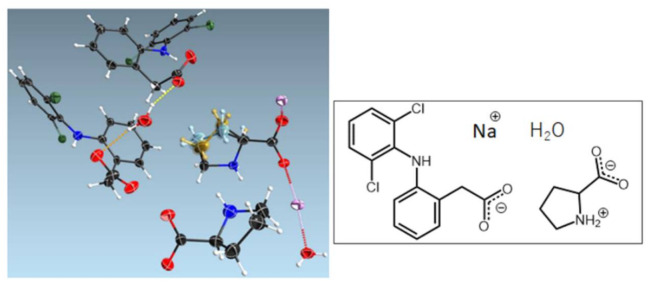
Crystal structure of sodium-diclofenac-l-proline tetrahydrate salt co-crystal with Na^+^ coordinated with two l-proline (LP) and four water molecules [22].

**Figure 7 molecules-26-03279-f007:**
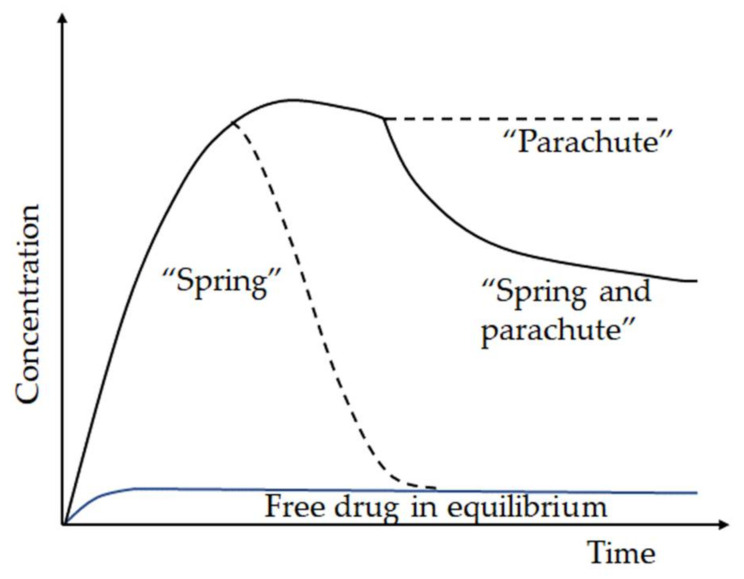
Schematic representation of API dissolution illustrating the “spring and parachute” model.

**Figure 8 molecules-26-03279-f008:**
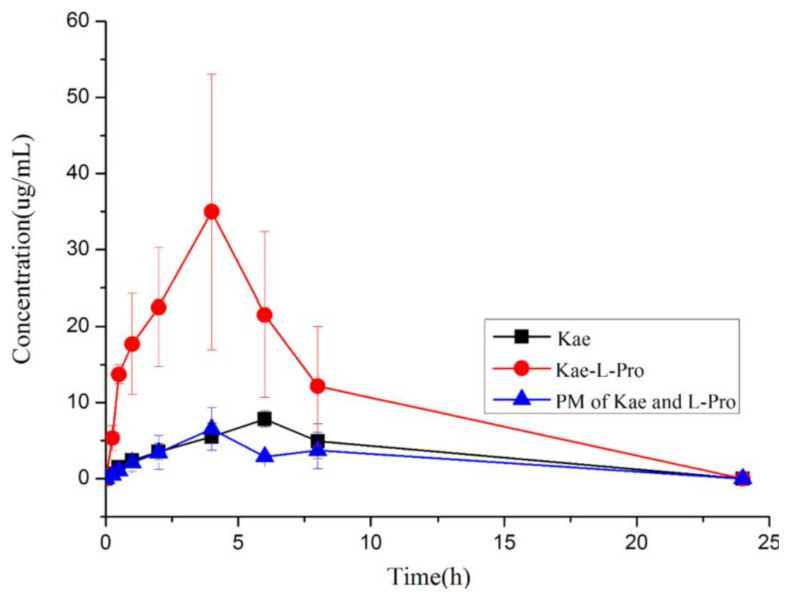
Pharmacokinetic profiles of pure kaempferol, kaempferol-l-proline co-crystal, and the physical mixture (Adapted from [103]).

**Table 1 molecules-26-03279-t001:** Amino acid classification based on their side chains [52].

Non-Polar Amino Acids	Polar Amino Acids	Electrically Charged Amino Acids	
		Acidic	Basic
Glycine (GLY) 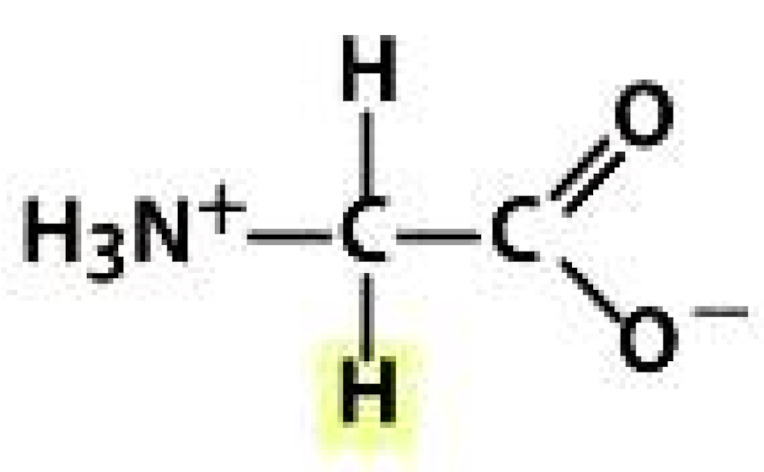	Serine (SER) 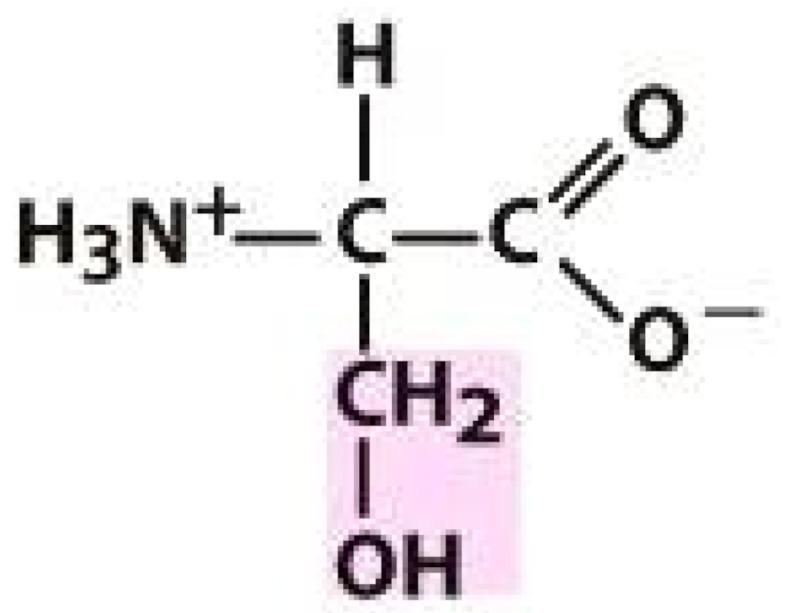	Aspartic acid (ASP) 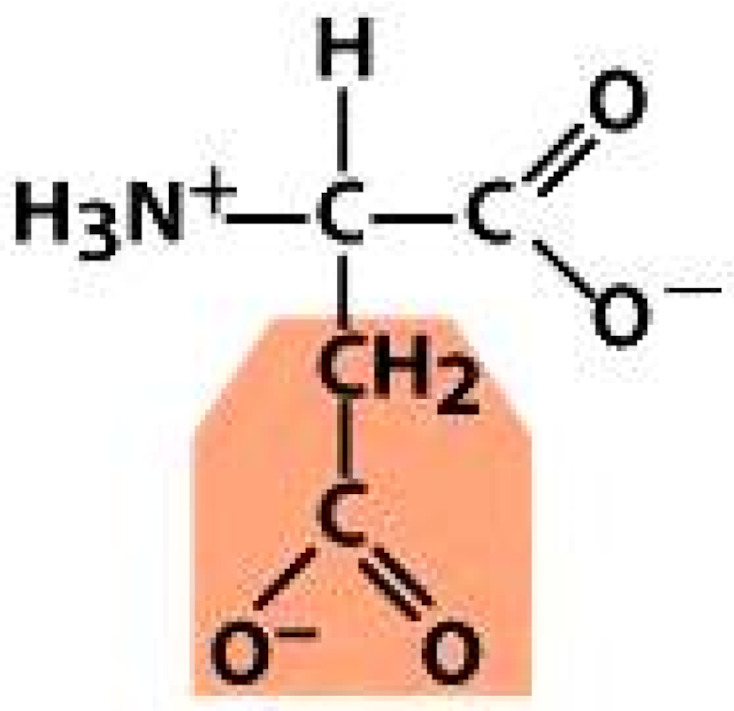	Histidine (HIS) 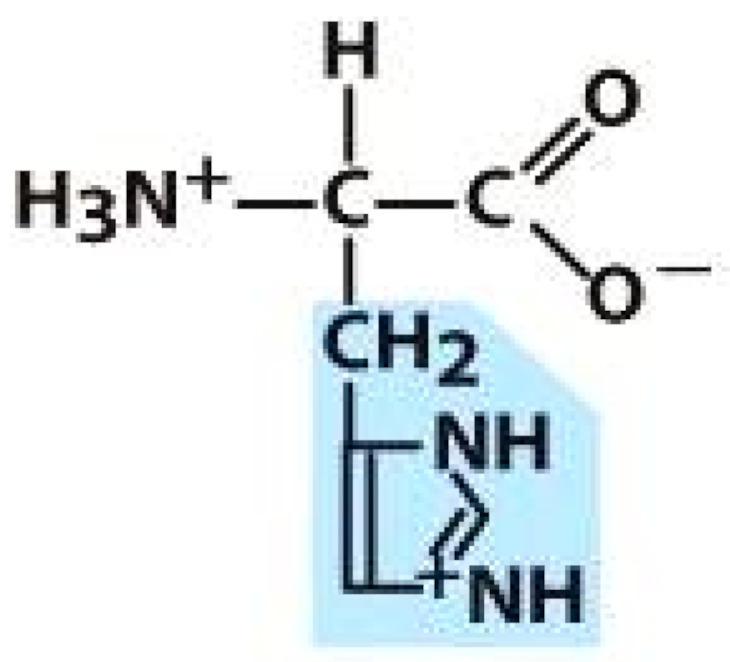
Alanine (ALA) 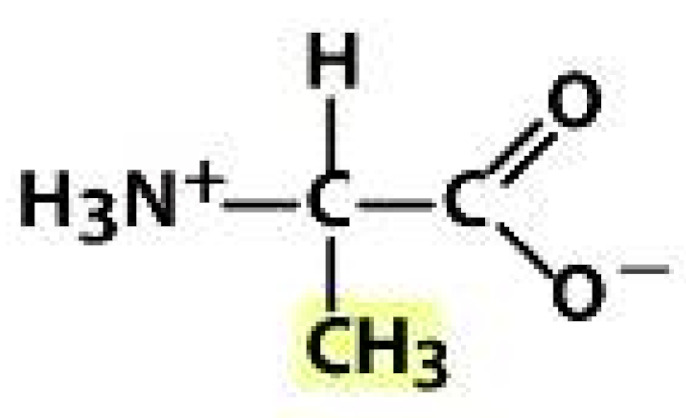	Threonine (THR) 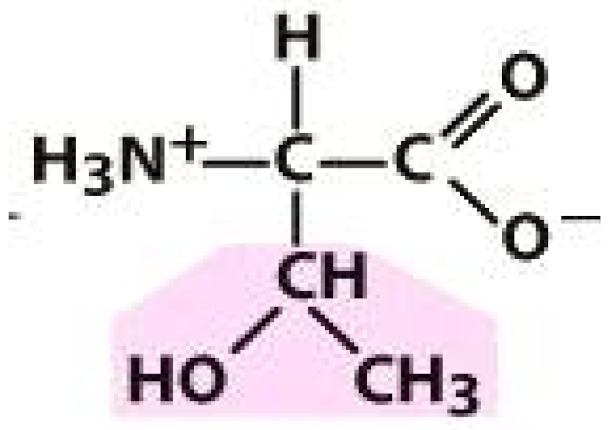	Glutamic acid (GLU) 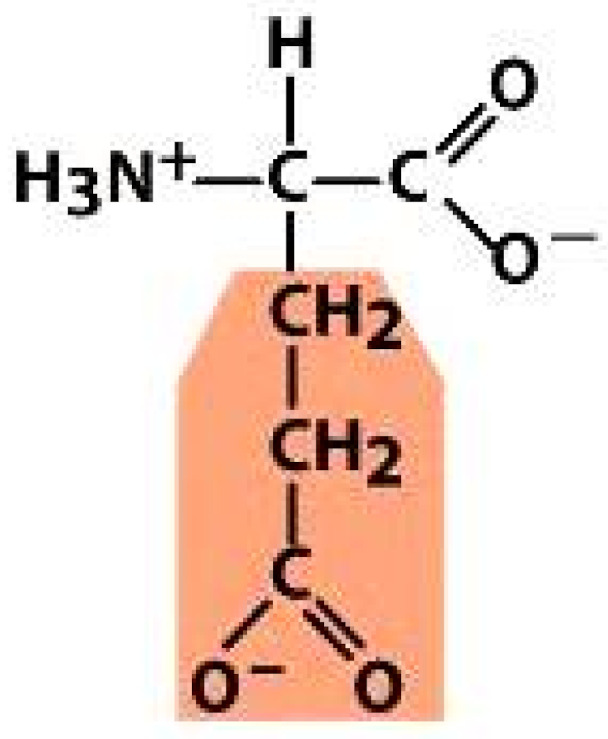	Arginine (ARG) 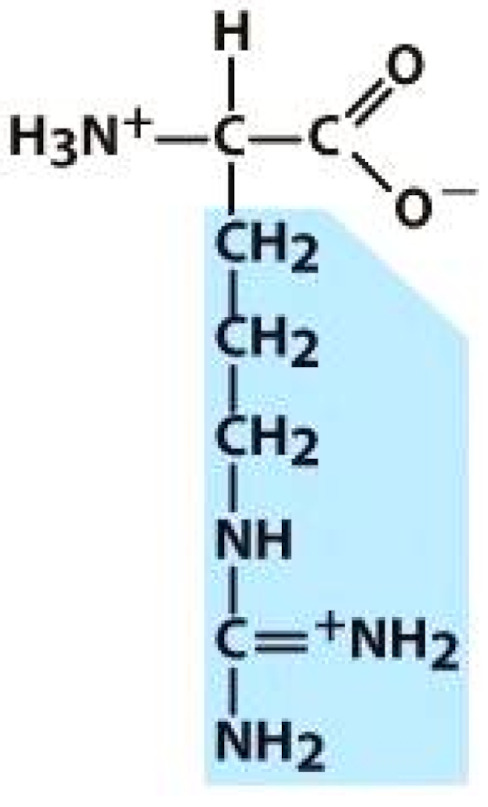
Valine (VAL) 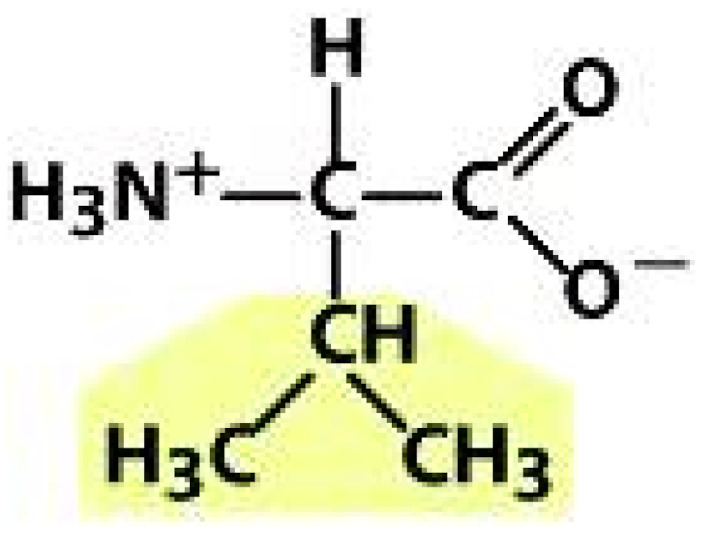	Cysteine (CYS) 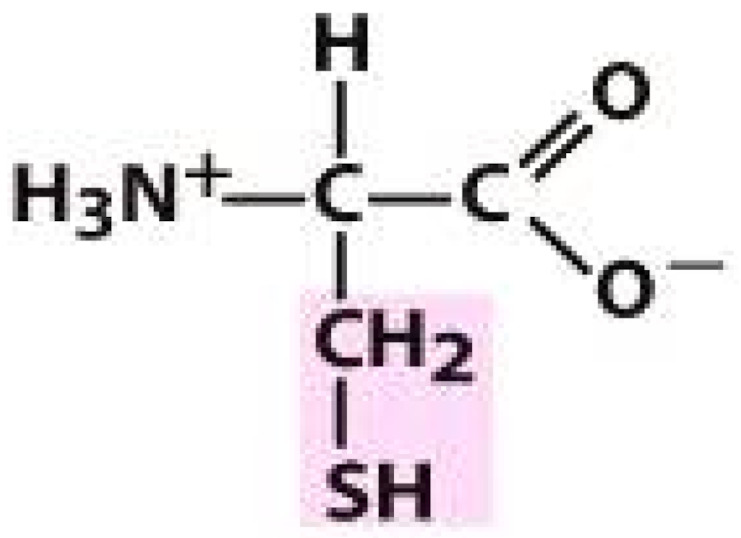		Lysine (LYS) 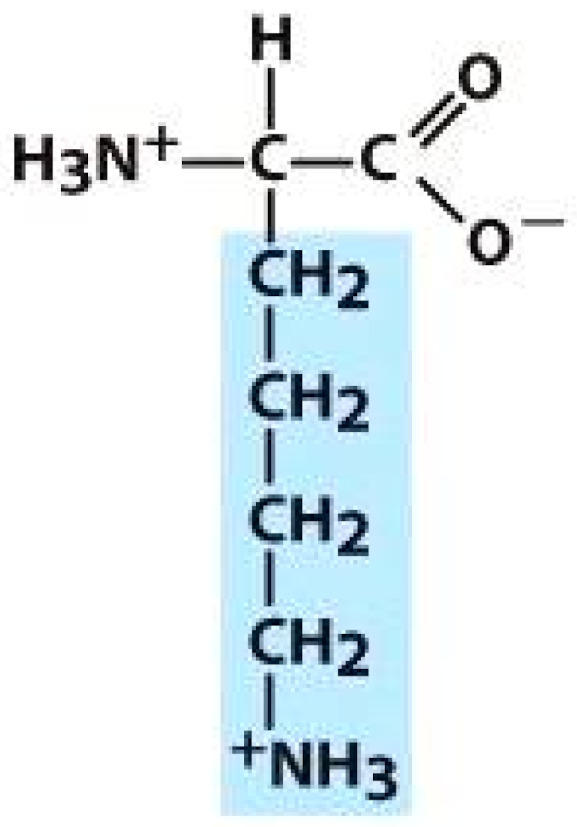
Leucine (LEU) 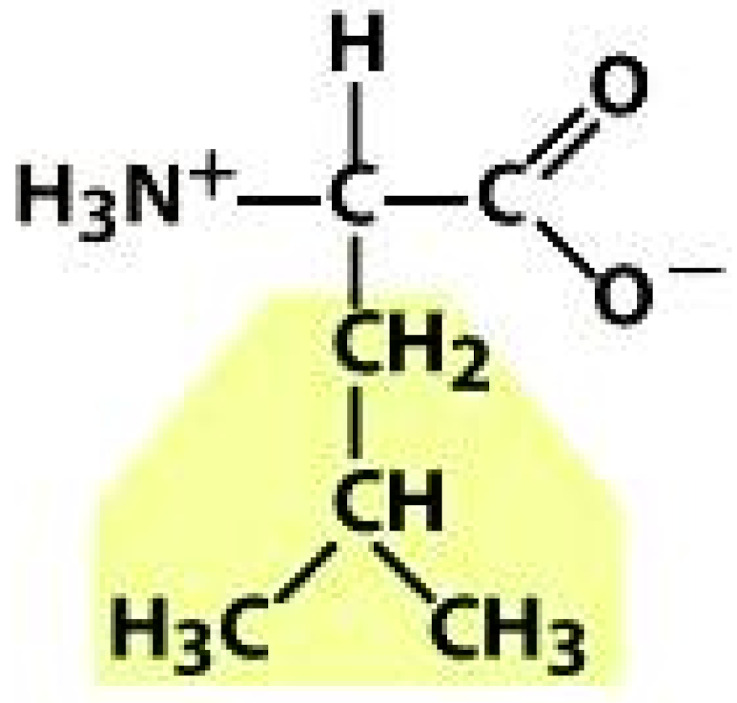	Tyrosine (TYR) 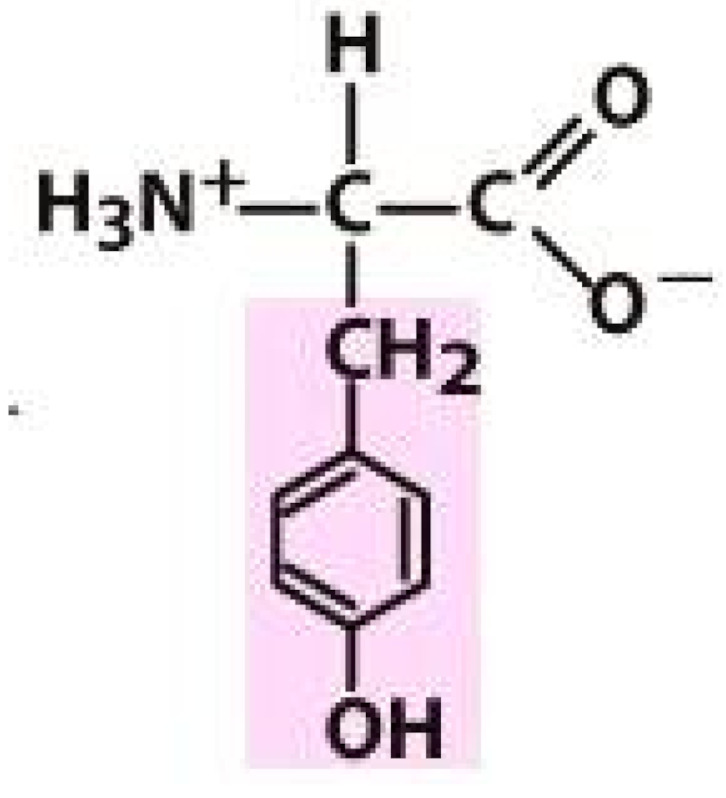		
Isoleucine (ILE) 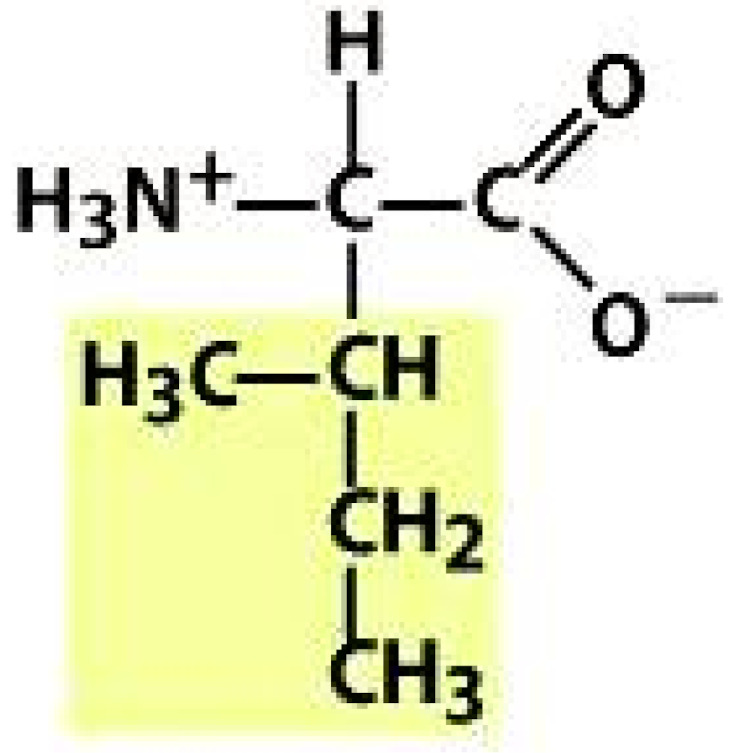	Asparagine (ASN) 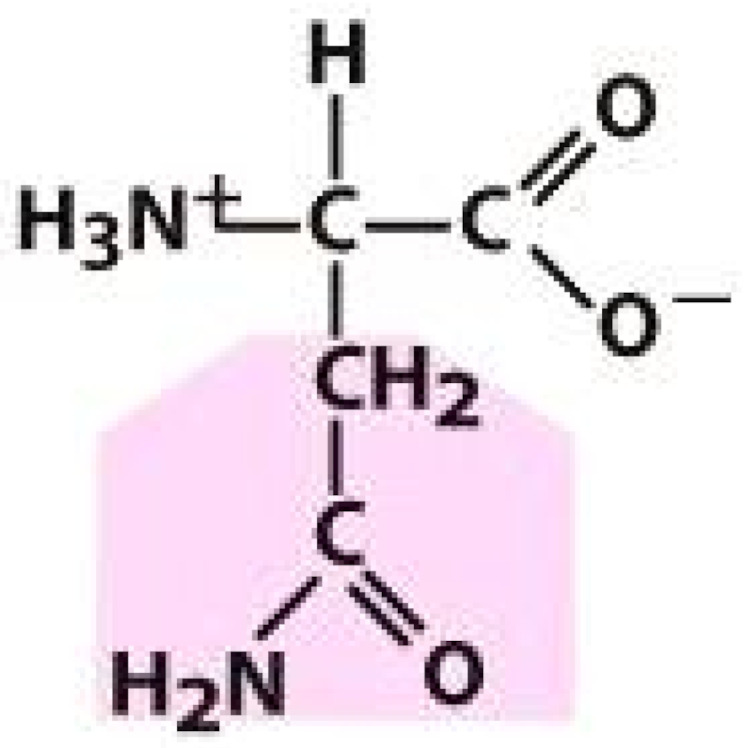		
Proline (PRO) 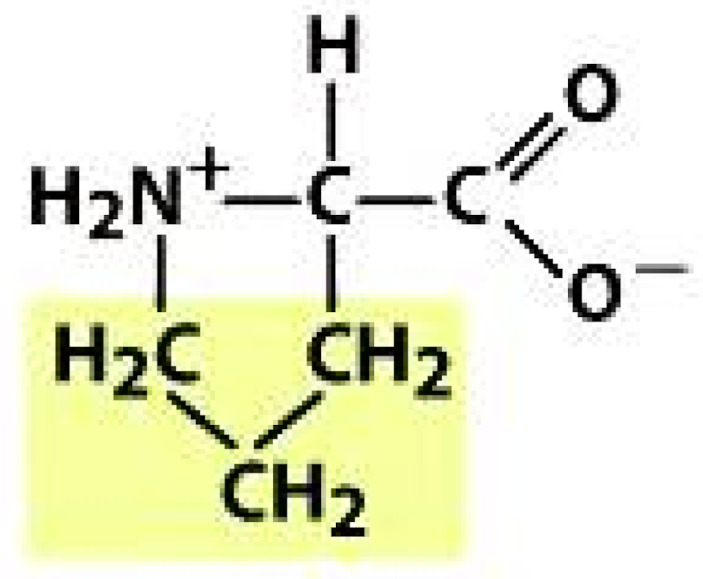	Glutamine (GLN) 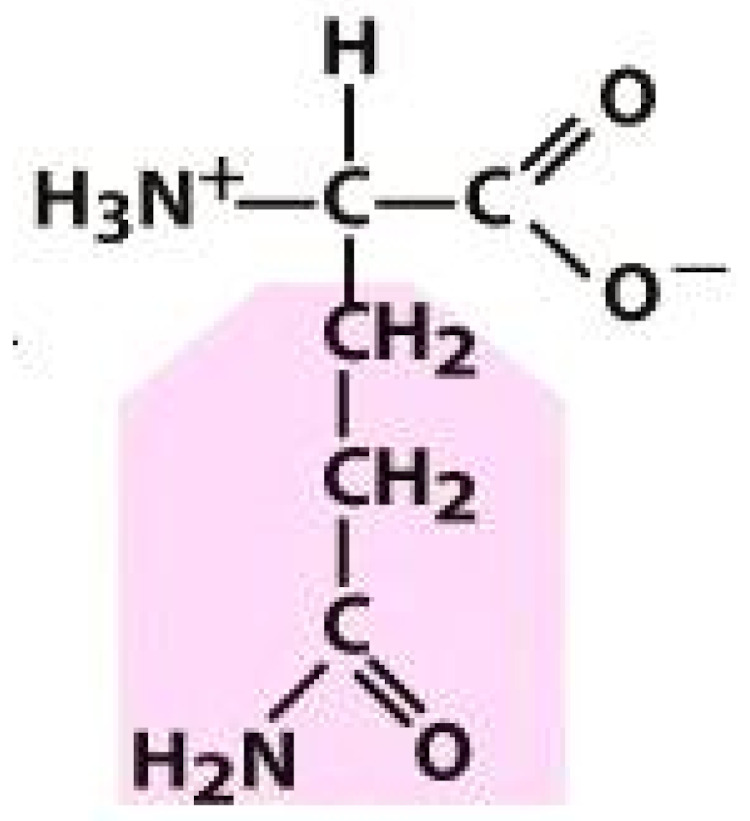		
Phenylalanine (PHE) 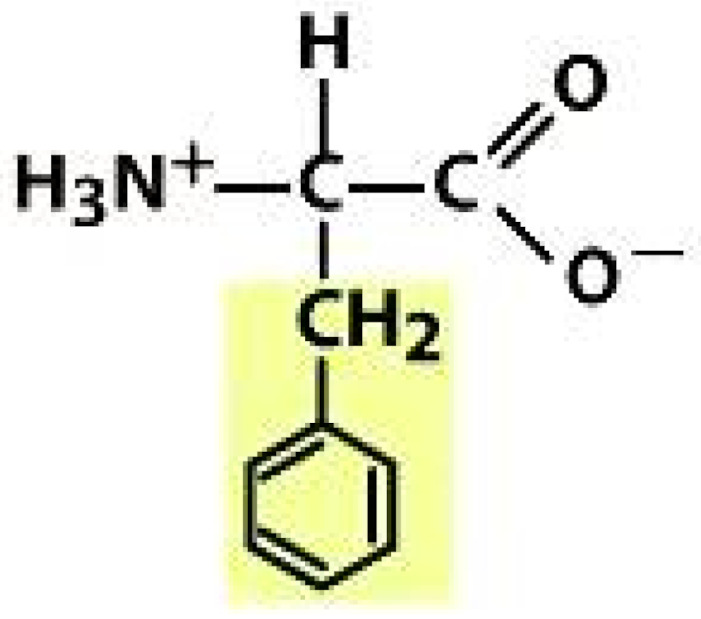			
Tryptophan (TRP) 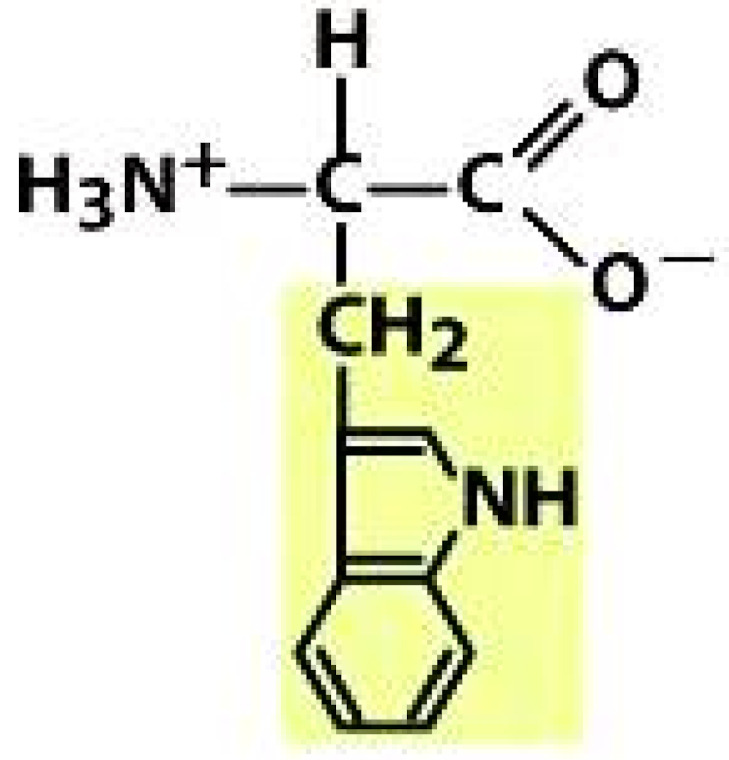			
Methionine (MET) 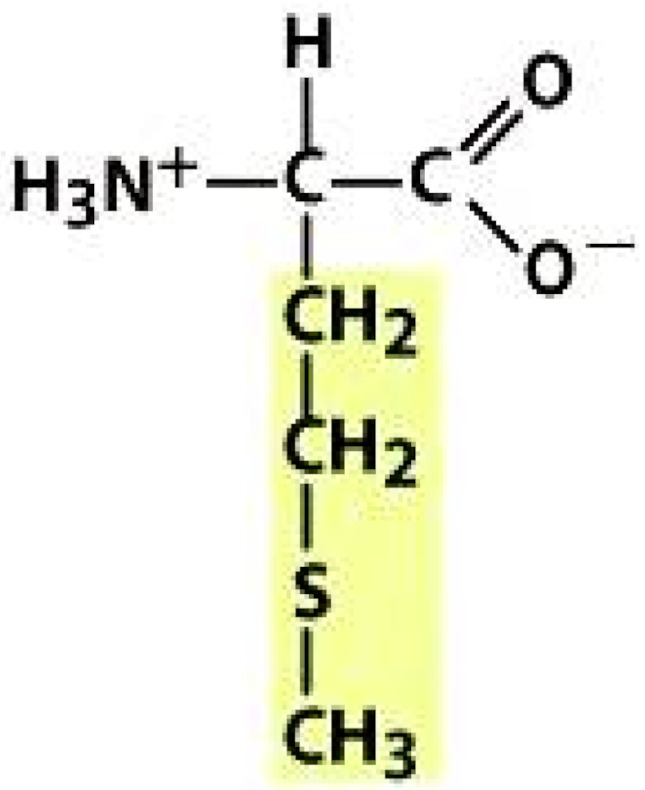			

**Table 2 molecules-26-03279-t002:** Summary of a few reports on API-amino acid co-crystals and the pharmaceutical improvement.

No	API-Amino Acid Co-Crystal	Molar Ratio	Preparation Method	Solubility	Dissolution	Pharmacokinetics Effect	Reference
1	Febuxostat-l-Pyroglutamic acid (FB-PG)	2:1	LAG ^1^	Co-crystal exhibits higher solubility than pure febuxostat in various medium	-	-	[12]
2	Myricetin-Proline	1:2	SE ^2^	Co-crystal exhibits higher solubility than pure myricetin and the physical mixture	Co-crystal exhibits a higher dissolution rate and increases the concentration of myricetin dissolved than pure myricetin and the physical mixture	Co-crystal exhibits faster absorption rate and higher Cmax, AUC value, and the relative bioavailability than pure myricetin	[19,104]
3	Acetazolamide-l-Proline (ACA-PRO)	1:1	SE ^2^	Co-crystal exhibits higher solubility than pure acetazolamide and the physical mixture in the three buffers tested. The solubility enhances with increasing pH values	The IDR values of the co-crystal in the three buffers are around three to four times over pure acetazolamide. The time that the co-crystal reached the maximum concentrations decreases with increasing pH values	Co-crystal exhibits higher Cmax and AUC value. Co-crystal showed a faster onset time and a longer duration of action than pure acetazolamide	[68]
4	Ritonavir-D-Alanine (RTN-DAL)	1:5	LAG ^1^	Co-crystal exhibits higher solubility than pure ritonavir, but not higher than ritonavir-succinic acid and ritonavir-adipic acid	Co-crystal exhibit marginal increase than pure ritonavir	-	[75]
5	Itraconazole-Aspartic acid	1:1	NG ^3^	Co-crystal significantly improves the solubility of itraconazole in simulated gastric fluid	Co-crystal showed enhancement in the dissolution rate than pure itraconazole	-	[78]
6	Itraconazole-Glycine						
7	Itraconazole-Proline						
8	Itraconazole-Serine						
9	Diclofenac sodium-l-Proline Tetrahydrate (NDPT)	1:1	SE ^2^	Co-crystal exhibits higher solubility than pure diclofenac sodium	Co-crystal exhibits a higher dissolution rate and increases the percentage of drug release than pure diclofenac sodium	-	[22]
10	Diclofenac sodium-l-Proline Monohydrate (NDPM)						
11	Indomethacin-l-Proline (IND-PL)	3:1	LAG ^1^	Co-crystal exhibits higher solubility than pure indomethacin and the physical mixture in the three buffers tested. The solubility enhances with increasing pH values	The IDR values of the co-crystal in the three buffers are around two times over pure indomethacin. The IDR values become higher with increasing pH values	Co-crystal showed a faster onset time and a longer duration of action than pure indomethacin. Co-crystal has comparably high bioavailability	[90]
12	Diclofenac-l-Proline (DFA-PRO)	1:1	LAG ^1^	Co-crystal exhibits higher solubility than pure diclofenac	-	-	[92]
13	Diclofenac Acid-Proline nano-co-crystal	1:1	Fast evaporation	-	The result indicates a 1.32-fold increase in nano co-crystal in pH 1.2 buffer, 1.14-fold in pH 6.8 buffer, and 2.46-fold in pH 7.4 buffer	-	[106]
14	Diclofenac potassium-l-Proline (DKPH)	1:1	SE ^2^	Co-crystal exhibits higher solubility than pure diclofenac potassium and the physical mixture	The IDR values of the co-crystal in two buffers are around three times over pure diclofenac potassium and the physical mixture	-	[85]
15	Chlorothiazide-dl-Proline (CTZ-DL-PRO)	1:1	LAG ^1^	-	Co-crystal showed a slight improvement in dissolution property	-	[96]
16	Ezetimibe-Proline (EZT-PRO)	1:1	SE ^2^	Co-crystal exhibits higher solubility than pure ezetimibe	Co-crystal exhibits a higher dissolution rate and increases the concentration of ezetimibe dissolved than pure ezetimibe	-	[101]
17	Kaempferol-l-Proline (Kae-l-Pro)	1:2	SE ^2^	Co-crystal exhibits higher solubility than pure kaempferol	Co-crystal exhibits a higher dissolution rate than pure kaempferol	Co-crystal exhibits higher Cmax and AUC value. Co-crystal showed a faster onset time than pure kaempferol and the physical mixture	[103]

Note:  ^1^ LAG: liquid assisted grinding; ^2^: solvent evaporation; ^3^ NG: neat grinding.

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
