# Peer review of "Amino Acids as the Potential Co-Former for Co-Crystal Development: A Review"

_molecules, 2021, doi:10.3390/molecules26113279_

Round 1

Reviewer 1 Report

The manuscript is a well written review paper of significant importance in the field. The authors review the potential of amino acids as co-formers for co-crystal development. There are a few points I would like to rise that might help to further improve the paper.

  • The introduction generally introduces into the subject but the aim of the paper is not stated.
  • Chapter 4 as the main chapter of the paper is very comprehensive in text and the "internal structure" of the section is not clear. Dividing it into suitable sub-sections should improve readability and structure.
  • page 10 2nd paragraph: "The thermogram of a physical mixture shows 2 endothermic peaks, each corresponding to the pure compound's melting point,...":  This is wrong; a physical mixture leads to a eutectic system. Therefore, the situation shown in Figure 4d is not existing at all.
  • For a review paper, the conclusions are a bit to focused on one amino acid as co-former. Also an outlook would be supportive at this point.

Some minor issues:

  • The right allocation of references has to be checked at some places, e.g.  2nd paragraph of page 11.
  • Figure 3 requires a reference ([67]?).
  • Table 2: Numbers 1-3 in column 3 are not described.

Reviewer 2 Report

If this is to be a review on "amino acids as the potential co-foremr for co-crystal ..." it cannot overlook entirely the whole class of aminoacid cocrystals with inorganic salts. Ionic co-crystals are mentioned on passing in the Introduction and not explored any further. If there is a reason for not reviewing this relevant branch of the co-crystals with aminoacids, the authors should say so otherwise the review is misleading.

Reviewer 3 Report

The review is very well written and represents a significant contribution in the field of research of co-formers for modifying the physicochemical properties of active pharmaceutical ingredients. The review definitely deserves to be published in Molecules. I only suggest a few minor language corrections:

In abstract, the sentence "Amino acids are a promising candidate due as they have functional groups that can form hydrogen bonds and are able to increase stability through zwitterionic moieties, which support strong interactions. " should be changed as "Amino acids are a promising candidate as they have functional groups that can form hydrogen bonds and are able to increase stability through zwitterionic moieties, which support strong interactions."

Page 3: in "(a) mostly all suitable proton donors (such as -COOH and -NH4+)" the "+" sign should be put in superscript.

Page 4: in "with charged side chains (Table 1) [46]." there are extra spaces between "(" and "Table 1)".

Page 6: in "the amino group in the amino acid is protonated so that it be-comes positively charged (-NH3+)." - "3" should be put as subscript.

Page 6: in "In contrast, the carboxylic group is deprotonated so that it becomes negatively charged (-COO-)." - "-" should be put as superscript.

Page 13: in "Lithium is a monovalent compound" the change should be made to "monovalent metal".

Round 2

Reviewer 2 Report

If this were a scientific paper reporting new findings there would be no problem in confining the discussion to a selected subset of systems but this is a review of aminoacid cocrystals and should be comprehensive and should not leave out recent literature. Papers by Shemchuk et al. in CGD and CEC on ionic cocrystals of proline and histidine and other aminoacids ought to be cited. 

Author Response

Dear Reviewer 2,

Thank you so much for your review and suggestion.

We have added four references about ionic co-crystal in this revision, that are reference number 15 -18, and then elaborated those references in abstract, introduction, discussion, and conclusion.

Abstract

We have improved the abstract and keywords to clarify and added a sentence in lines 14-15 regarding the reference changes.

Co-crystals are one of the most popular ways to modify the physicochemical properties of active pharmaceutical ingredients (API) without changing pharmacological activity through non-covalent interactions with one or more co-formers. “Green method” has recently prompted many researchers to develop solvent-free techniques or minimize solvents for arranging the eco-friendlier process of co-crystallization. Researchers have also been looking for less-risk co-formers that produce the desired API’s physicochemical properties. This review purposed to collect the report studies of amino acids as the safe co-former and explored their co-crystal advantages. Structurally, amino acids are the promising candidates as they have functional groups that can form hydrogen bonds and increase stability through zwitterionic moieties, which support strong interactions. The co-crystals yielded from this natural compound have been proven can improve pharmaceutical performance. For example, L-glutamine could reduce the side effects of mesalamine through an acid-base stabilizing effect in the gastrointestinal fluid. Moreover, some amino acids, especially L-proline, enhances API’s solubility and absorption both in the neutral and salt co-crystals forms. In addition, some ionic co-crystals of amino acids have also been designed to increase chiral resolution. Therefore, amino acids are safe potential co-formers for co-crystal development, which suitable for improving the physicochemical properties of API and prospective to be developed further in the dosage formula.”

Keywords: co-crystal; anionic co-crystal; salt co-crystal; amino acids; L-proline

Introduction (page 1, paragraph 4)

“Recently, a new category of multicomponent crystal, namely ionic co-crystal, is taking much attention due to its advantages, such as the simplicity and functionality to improve physicochemical properties of pharmaceuticals, food and fertilizers, and chiral resolution [15]. In that development, amino acids become the main co-former, i.e., in the ionic co-crystal formation of levodopa with LiCl and L-tyrosine and L-phenylalanine as the biological precursors [16]. Besides, the hydrated ionic co‐crystals from enantiopure L‐proline and racemic DL‐proline with LiX (X=Cl, Br, I) were also reported can increase chiral resolution [17], as well as DL-amino acids alanine, valine, leucine, and isoleucine with LiCl [18].”

Section 4.2.1 The Structure of L-proline-based Co-crystals; page 13, paragraph 3

“Ionic co-crystals of LiX (X=Cl, Br, I) with L-proline and DL-proline were reported. Those solid phases consisted of the inorganic element with this amino acid formed conglomerates (with Cl and Br) and racemates (with Cl and I), which produced distinct crystal layers between the opposite chirality. Hence, this developed method offers an advantage to chiral resolution [17].”

Conclusion (line-8)

We revised the conclusion to clarify and added the information considering the new references about ionic co-crystal (line 8-9, 10).

Amino acids are potential co-formers for supporting the “green method” co-crystallization of API. Co-crystals formed by API and amino acid have been shown to improve the drug’s physicochemical properties, such as solubility, dissolution rate, bioavailability, as well as chemical and physical stability. Structurally, amino acids in a co-crystal are in their zwitterionic form and produce a head-to-tail charge-assisted hydro-gen-bonded chain with the API, building strong interactions and enhance stability. L-proline exhibits the highest capacity to constructs co-crystal and modulates the pharmaceutical performance due to its rigid zwitterionic structure, solubility, and hydrotropic activity. In addition, some ionic co-crystals from amino acids include L-proline, also were developed to separate their chiral mixture. Hereafter, amino acid co-formers, especially L-proline, can be explored further for superior dosage forms development and chiral resolution methods in a simple, safe, economist, and eco-friendly way.

We do hope this improvement fulfills your suggestion.

Thank you.

Best regards,

Ilma Nugrahani
